# Exploiting the tunability of stimulated emission depletion microscopy for super-resolution imaging of nuclear structures

Maria J. Sarmento [1,5], Michele Oneto[1], Simone Pelicci[1,2], Luca Pesce[1,2], Lorenzo Scipioni [1], Mario Faretta[3], Laura Furia[3], Gaetano Ivan Dellino[3,4], Pier Giuseppe Pelicci[3,4], Paolo Bianchini [1], Alberto Diaspro [1,2] & Luca Lanzanò [1]

Imaging of nuclear structures within intact eukaryotic nuclei is imperative to understand the effect of chromatin folding on genome function. Recent developments of super-resolution fluorescence microscopy techniques combine high specificity, sensitivity, and less-invasive sample preparation procedures with the sub-diffraction spatial resolution required to image chromatin at the nanoscale. Here, we present a method to enhance the spatial resolution of a stimulated-emission depletion (STED) microscope based only on the modulation of the STED intensity during the acquisition of a STED image. This modulation induces spatially encoded variations of the fluorescence emission that can be visualized in the phasor plot and used to improve and quantify the effective spatial resolution of the STED image. We show that the method can be used to remove direct excitation by the STED beam and perform dual color imaging. We apply this method to the visualization of transcription and replication foci within intact nuclei of eukaryotic cells.

[1] Nanoscopy, Nanophysics, Istituto Italiano di Tecnologia, via Morego 30, 16163 Genoa, Italy. [2] Department of Physics, University of Genoa, via Dodecaneso 33, 16146 Genoa, Italy. [3] Department of Experimental Oncology, European Institute of Oncology, Via Adamello 16, 20139 Milan, Italy. [4] Department of Oncology and Hemato-Oncology, University of Milan, Via Santa Sofia 9, 20142 Milan, Italy. [5]Present address: Department of Biophysical Chemistry, J. Heyrovský Institute of Physical Chemistry of the A.S.C.R. v.v.i., Prague, Czech Republic. Correspondence and requests for materials should be addressed to A.D. (email: alberto.diaspro@iit.it) or to L.Lò. (email: Luca.Lanzano@iit.it)

The overall function of DNA in cell nuclei has long been related with different levels of genome spatial organization. Proper chromatin structure and dynamics are suggested to play an active role in gene regulation and consequently to be required for healthy cell proliferation and maintenance. For this reason, imaging/mapping nuclear structures within intact eukaryotic nuclei is imperative to understand the effect of chromatin structure on genome function. An example is represented by the study of essential nuclear functions such as DNA transcription and replication, both occurring in the context of highly structured chromatin. Since replication forks, while progressing throughout the genome, compete for the DNA template with active RNA polymerases, these two processes must be tightly coordinated in time and space (i.e., separation in different nuclear territories) to ensure proper execution. Nonetheless transcription–replication conflicts represent both in eukaryotes and prokaryotes a major source of spontaneous genomic instability, an hallmark of cancer cells[1]. Genome-wide analyses of replication and transcription by next-generation sequencing (i.e., ChIP-seq, Repli-seq, etc.) can provide high-resolution mapping of these potential collision hotspots[2,3]. However, most of these approaches measure population averages, thus failing to detect events that occur at the same time, and in the same place, within a minority of cells. Imaging and nanoscopy can thus provide a unique view of chromatin organization and structure in intact cell nuclei.

In the past, both chromatin structure size and its high level of compaction rendered electron microscopy (EM) as the method of choice to visualize the genome spatial organization. Its high resolution (~nm) allowed the study of chromatin at the nanoscale[4–8]. However, EM lacks the molecular specificity required to provide information on the identity of molecules composing macromolecular complexes and the harsh sample preparation can not only introduce some structural artifacts[9–11] but also preclude its use on living cells. Far-field fluorescence microscopy on the other hand overcomes these limitations, being compatible with the less-invasive labeling of specific molecules and their imaging within living cells. Yet, these improvements are achieved at the expense of resolution, which is limited by light diffraction to 200–300 nm.

The recent development of the so-called super-resolution fluorescence microscopy (SRM) techniques is filling the gap between these two approaches, by combining high specificity, sensitivity, and less-invasive sample preparation procedures with sub-diffraction spatial resolution (1–100 nm). For this reason, SRM methods are well suited to study chromatin spatial arrangement at the nanoscale within intact nuclei. So far, several SRM techniques were applied to the nuclear compartment, including distinct approaches based on structured illumination microscopy[12–14], single-molecule localization microscopy[15–18] and, in less extent, stimulated emission depletion (STED)[19,20]. For example, RNA polymerase II (RNApolII) organization was visualized by two-dimensional single-particle tracking combined with photoactivation localization microscopy[21] and by stochastic optical reconstruction microscopy (STORM) in reflected light-sheet configuration[22]. Both studies show the transient character of RNApolII clustering, therefore contradicting the presence of stable preassembled transcription foci. Another example is the study of endogenous H2B histone by STORM[23], in which the authors show that nucleosomes are arranged in heterogeneous "clutches" separated by nucleosome-free regions. In both cases, the use of SRM resulted in new and vital information regarding subnuclear organization that was previously impossible to acquire with diffraction-limited approaches. Many other studies comprising the direct in situ visualization of not only DNA-binding proteins but also the DNA itself are now shedding some new light into nuclear organization and function[24–27].

More recently, our group introduced a novel approach to achieve the nanoscale resolution required to image subnuclear structures: separation of photons by lifetime tuning (SPLIT)[28]. In this approach, the phasor plot is used to decode spatial information encoded in an additional channel and extract components of different spatial resolution[28]. For instance, in the specific implementation described in ref.[28], a STED beam was used to generate a fluorescence lifetime gradient across the detection volume and the phasor plot was applied to decode the spatial information encoded within this lifetime gradient. This resulted in an improvement of the spatial resolution of the STED image with simultaneous removal of STED-induced background[28]. In that specific implementation, the generation and observation of fluorescence lifetime gradients required pulsed excitation and dedicated hardware for lifetime detection in the nanosecond temporal scale, which is not available on every STED microscope. However, it is worth noting that the SPLIT approach is more general and the only prerequisite to its application is the presence of encoded spatial information in an additional channel. This additional channel can be represented by any of the variables available in the multidimensional optical microscope[29]. In particular, a fundamental property of STED, compared to other SRM techniques, is that its spatial resolution is tunable as a function of the depletion power. This unique property has been largely exploited in STED-based spot-variation fluorescence correlation spectroscopy (FCS) for measuring molecular diffusion at different spatial scales[30,31] but, to the best of our knowledge, has not been used to improve super-resolved imaging.

Here, we take advantage of this tunability of the STED microscope to encode spatial information within variations of the depletion power and then decode this information by SPLIT. We show that an improvement in resolution can be achieved by modulating the STED power and analyzing the encoded spatial information using phasors. An obvious advantage of this approach is that it can be readily used on any STED microscope, including those that are commercially available. We apply this methodology to the imaging of transcription and replication foci within intact nuclei of eukaryotic cells. We show that our analysis provides knowledge on the effective improvement of spatial resolution and that this information can be used to estimate the true size of transcription foci without prior calibration of the microscope Point Spread Function (PSF). Additionally, we show the applicability of this approach to the particular case of fluorophores presenting STED-induced background. Using the phasor plot, we are able to separate the background and retrieve images with increased resolution, which would be masked otherwise. We additionally exploit the background subtraction to perform dual color imaging.

## Results

**Modulation-enhanced STED microscopy.** The improvement in resolution achieved with STED microscopy results from the ability to silence part of the fluorophores within the diffraction-limited PSF. This condition can be made less constrictive by encoding spatial information into an additional channel. In modulation-enhanced STED microscopy (M-STED), the information required for resolution improvement is encoded within the modulation of the STED power during the acquisition of a STED image. In any STED microscope, the fluorescence signal $i(r)$ originating from a molecule located at distance $r$ from the center of the PSF can be well approximated by[32]

$$i(r) = i(0)e^{-\frac{2r^2}{w_c^2}}e^{-\frac{I_{STED}(r)}{I_{sat}}},\qquad(1)$$

where $w_c$ is the waist of the confocal PSF, $I_{sat}$ is a constant, and $I_{STED}(r) = I_w \cdot r^2/w_c^2$ represents a parabolic approximation for a

doughnut-shaped STED beam, with $I_w$ being the STED beam intensity at position $r = w_c$. Within this approximation, the effective waist of the STED PSF, $w$, is given by

$$\frac{1}{w^2} = \frac{1}{w_c^2}\left(1 + \frac{1}{2}\frac{I_w}{I_{sat}}\right) \tag{2}$$

showing that the resolution of a STED microscope is tunable as a function of the depletion (STED laser) power.

Now, if the STED intensity is modulated in time, for instance is increased linearly from 0 to a value $I_{max}$ during a time interval $T$, $I_w(t) = I_{max} \cdot t/T$ (Supplementary Fig. 1a), then the fluorescence signal will also be temporally modulated in a spatially dependent manner:

$$i(r,t) = i(0)e^{-\frac{2r^2}{w_c^2}}e^{-\frac{I_{max}}{I_{sat}}\frac{r^2}{w_c^2}\frac{t}{T}}. \tag{3}$$

Notably, Eq. (3) represents an exponential decay with a spatially dependent decay constant $\gamma(r) = (I_{max}/I_{sat})(r^2/w_c^2)/T$. In other words, the signal from fluorophores located in different parts of the PSF will show intensity variations in the temporal channel that are function of the fluorophore's position within the PSF (Fig. 1a, b). In the center, fluorescence emission remains constant (Fig. 1b, in). Away from the center, fluorescence emission decreases in a manner that is intrinsically dependent on the STED modulation. In formal analogy to ref.[28], we may refer to these variations as "fluorescence depletion dynamics". However, while in ref.[28] the word dynamics was indeed used to indicate the nanosecond fluorescence lifetime, here it denotes the intensity variations in response to a modulation of the STED

power that can be implemented at an arbitrary timescale. The fluorophore's nanometer spatial distribution can then be assessed by applying the SPLIT method to analyze these variations. In the phasor representation, $P_{in}$ and $P_{out}$ are the phasors corresponding to the center and the periphery of the PSF (Fig. 1c), respectively. This difference between center and periphery can be exploited to apply the SPLIT method and further enhance the spatial resolution of the image (Fig. 1d).

The application of the method to simulated images of nuclear foci is shown on Fig. 1e–i. A series of $n = 8$ images was generated to simulate increasing STED power throughout the stack (Fig. 1e), from the confocal image $F_1$ ($P_{STED} = 0$) to the one with the highest STED power and therefore the best spatial resolution ($F_8$, $P_{STED} = P_{max}$). To create such image stack, point-like objects were convolved with the theoretical PSF of a STED microscope described by Eq. (1), with STED intensity $I_w$ varying from 0 in the first frame to $I_{max}$ in the last frame. The corresponding phasor spreads away from the origin, highlighting the heterogeneity of the introduced depletion dynamics (Fig. 1f, see for comparison Supplementary Fig. 2 showing the phasor plot of a series of images acquired at constant STED power). Separating the fluorescence emission contribution from the center of the PSF ($P_{in}$) from the one originating from the periphery ($P_{out}$) using the SPLIT algorithm (Fig. 1h) gives rise to an image $F_{in}$ of improved resolution (hereafter referred to as the SPLIT image), when compared with $F_8$. Fitting the line profiles in Fig. 1i with a multi-peak Gaussian function (not shown), we estimate the respective full-width at half-maximum (FWHM) of these particular features to be ~130 and ~114 nm for the STED image $F_8$ and ~97 and ~88 nm for the SPLIT image. This represents an improvement in resolution in this particular simulation of around 30 nm. Note

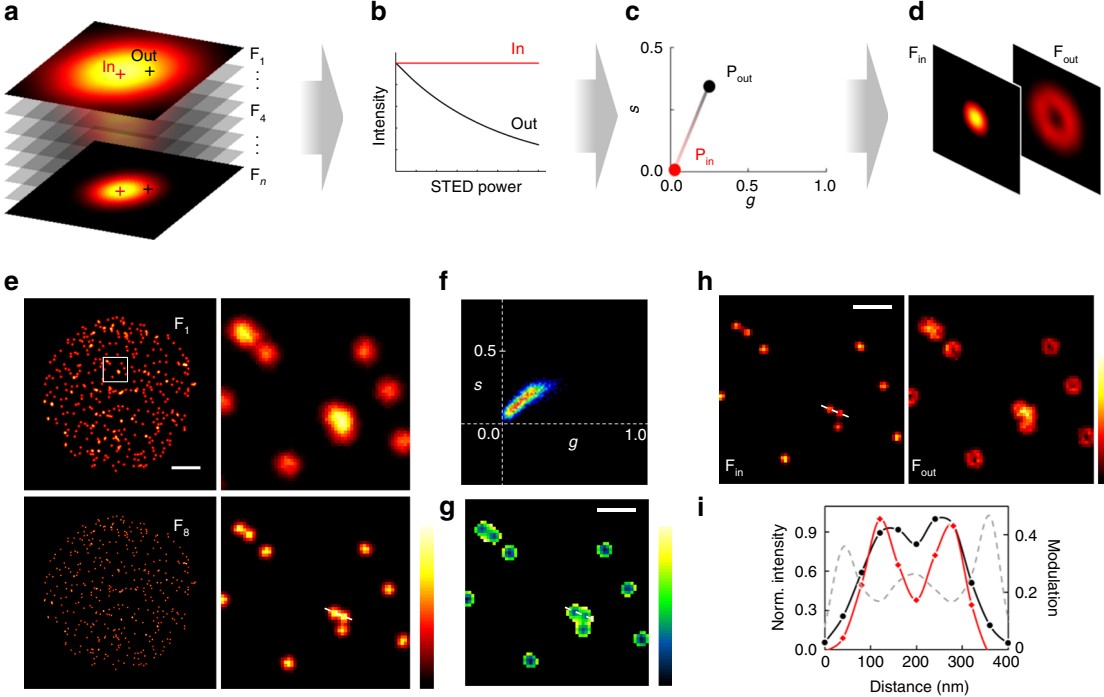

**Fig. 1** M-STED microscopy implementation. **a** Schematic representation of acquired image stack with increasing STED power, from $F_1$ to $F_n$. **b** Fluorescence depletion dynamics of photons arising from the center (in) and the periphery (out) of the PSF, along the stack. **c** Phasor representation of the dynamics depicted in **b**. **d** Example of images obtained upon application of the SPLIT methodology. **e** Simulated M-STED imaging of nuclear foci. The maximum saturation level was set to $I_{max}/I_{sat} = 8$. Color scale: normalized intensity. Scale bar: 3 μm. **f** Phasor representation of the depletion dynamics in **e**. **g** Modulation image $M(x,y)$ encoding the spatial information. Color scale: modulation $M$ (0–0.55). Scale bar: 500 nm. **h** SPLIT image components resulting from the efficient separation of fluorescence emission arising from the center $F_{in}$ and the periphery $F_{out}$. Color scale: normalized intensity. Scale bar: 500 nm. **i** Comparison of the ability to separate two foci, and hence the resolution, between the $F_8$ (black circles) and $F_{in}$ (red diamonds), and the relation with the information encoded in $M(x,y)$

that while in ref.[28] the intrinsic fluorescence decay of the fluorophore (measured in the nanosecond timescale) was used as means to separate in and out originated photons, here the same goal is achieved by analyzing STED images of varying resolution. For any given pixel $(x,y)$, the fractions of the intensity corresponding to the center and the periphery of the PSF are obtained from the distance between the phasor measured in that pixel, $\mathbf{P}(x,y)$, and the phasors $\mathbf{P}_{out}$ and $\mathbf{P}_{in}$, respectively (Supplementary Fig. 3a). It is worth noting that, even though a phasor position is always described in polar coordinates by a modulation $M$ and a phase $\phi$ (Supplementary Fig. 3b), in this simulation only the former is encoding significant spatial information visible in the modulation image $M(x,y)$ (Fig. 1g). As expected, $M(x,y)$ presents minima at the center of the foci, and higher values towards their periphery. Note also that the SPLIT algorithm can be used to improve the spatial resolution of any of the STED images available in the stack. For instance, one can generate the SPLIT image using a sum of the images of the stack (Supplementary Fig. 4).

**Depletion curve and phasor analysis in M-STED microscopy.** The SPLIT separation of the signal at each pixel relies on a proper assignment of the positions of the phasors $\mathbf{P}_{in}$ and $\mathbf{P}_{out}$ corresponding to the center and periphery of the detection volume, respectively. In ref.[28] these positions were calculated based on the knowledge of the unperturbed fluorescence lifetime of the fluorophores and a parameter that quantifies the variation of decay rates within the PSF. This parameter was extracted by fitting the average intensity profile along the stack to an analytical model derived as the convolution of multiple exponential decays[28]. Here, a similar model can be used to describe the depletion curve, i.e., the average variation of fluorescence intensity as a function of the STED power (Supplementary Note 1):

$$\frac{\langle F(x,y,j)\rangle}{\langle F(x,y,1)\rangle} = \frac{1}{1 + \frac{k}{2}\frac{(j-1)}{(n-1)}}, \quad (4)$$

where the brackets denote averaging over the entire image and the constant $k = I_{max}/I_{sat}$ represents the saturation value at position $r = w_c$ for the image of the stack acquired at maximum STED power ($j = n$). Estimation of this value is important for at least two reasons. First, for any given sample, it provides direct knowledge on the improvement of resolution introduced by the M-STED PSF (Eq. (3)) with respect to the confocal PSF. For instance, according to Eq. (2), for $I_{max} = 8 \cdot I_{sat}$ the size of the PSF corresponding to the last frame of the stack is $1/(1 + 0.5 I_{max}/I_{sat})^{1/2} = 0.45$ times the size of the confocal PSF. Second, the parameter $k$ quantifies the spatial variation of the depletion dynamics within the PSF. For example, at a distance $r = w_c$ from the center of the PSF $\gamma(r) = (I_{max}/I_{sat})/T$, whereas in the center $\gamma(0) = 0$.

By simulating images using the M-STED PSF described by Eq. (3), we verified that the value of $k$ extracted from the fit of the depletion curve was always in keeping with the simulated value of $I_{max}/I_{sat}$ (Fig. 2), independently from the image SNR (Fig. 2a, b, Supplementary Fig. 5a) or the simulated object (Fig. 2a–d, e and Supplementary Fig. 5b, c). Importantly, the quantitative information on the STED saturation level, extracted here from the STED power-dependent intensity variations, cannot be obtained from a STED image acquired at a single STED power (Supplementary Fig. 2).

Simulations of sparse foci with the same SNR but different value of $I_{max}/I_{sat}$ show that a lower $k$ value corresponds to a smaller elongation of the phasor plot (Fig. 2a–c), indicating that the ability to separate the photons from the center and the periphery of the PSF is hampered. However, the value of $k$ does not tell us the minimum separation between $\mathbf{P}_{in}$ and $\mathbf{P}_{out}$ that we can actually resolve at the SNR available in the images. In order to overcome this issue, we introduce an intuitive method based on the graphical analysis of the phasor plot that is also sensitive to the SNR of the images. Figure 2a–c show simulations of sparse foci under different conditions of acquisition, namely different values of $I_{max}/I_{sat}$ and different levels of SNR. In the phasor plot, the effect of both $I_{max}/I_{sat}$ and SNR can be visually evaluated. The phasor elongation along the radial direction, $\Delta_M$, is mostly affected by the induced modulation and therefore the higher $I_{max}/I_{sat}$ the larger $\Delta_M$. The spread along the angular direction, $\Delta_N$, describes mainly the SNR, with higher SNR values resulting in smaller $\Delta_N$. From the combination of $\Delta_M$ and $\Delta_N$, one can estimate $M_0$ that ultimately represents the smallest difference in modulation between $\mathbf{P}_{in}$ and $\mathbf{P}_{out}$ that can be distinguished and hence used for SPLIT.

Ideally, the experimental conditions should result in a phasor plot with large $\Delta_M$ and small $\Delta_N$. Looking closer at the simulations, one can see that for the same $I_{max}/I_{sat}$ (Fig. 2a, b), a lower SNR results in a larger $\Delta_N$ and thus a larger $M_0$, and the resulting SPLIT image becomes less resolved. Similarly, a larger value of $M_0$ is also observed when the SNR is kept constant and $I_{max}/I_{sat}$ is decreased instead (Fig. 2a–c), resulting in a smaller $\Delta_M$ value. Nevertheless, the resolution of the final SPLIT image is much better in Fig. 2b than in Fig. 2c, despite a similar $M_0$, because it is generated from an $F_8$ image of higher spatial resolution. Finally, reducing both $I_{max}/I_{sat}$ and SNR results in a phasor with $\Delta_M \approx \Delta_N$ that prevents any improvement of resolution based on SPLIT (Supplementary Fig. 5a). In addition, we observed that for more crowded samples (Fig. 2d, e) there is a gap between the phasor plot and the origin, indicating that the theoretical depletion dynamics $\gamma = 0$, corresponding to the center of the PSF, is not represented in any pixel of the image. In other words, in crowded samples, the modulation is different from zero even in the center of the foci (where in principle $I_{STED} = 0$ and fluorescence is constant throughout the stack) because fluorescence coming from the periphery of other foci (and thus modulated) is also detected in every pixel. In this case, the best SPLIT separation is provided by setting $\mathbf{P}_{in}$ at a position corresponding to the minimum modulation observed in the data (Fig. 2d, e).

It is worth noting that the minimum number of frames required to calculate a modulation is $n = 2$. However, for $n = 2$ the value of the phasor coordinate $s$ is systematically null (see Eq. (7)) and thus we cannot take advantage of the graphical analysis of the phasor plot (Supplementary Fig. 6a). The minimum number of frames required to perform a graphical analysis of the phasor plot is $n = 3$ (Supplementary Fig. 6b, c). The use of larger values of $n$ allows the calculation of the phasor plot also at higher frequencies (Supplementary Fig. 6d).

**M-STED microscopy of transcription foci.** Next, we applied M-STED to the direct visualization of transcription foci within intact nuclei of eukaryotic cells. MCF10A cells were loaded with the nucleotide analog 5-bromouridine (BrU) that is incorporated into nascent RNA during transcription. Upon fixation, BrU was immunostained with anti-BrdU(rabbit)/anti-rabbit Chromeo488 as described in the Methods section. M-STED was then performed by acquiring eight frames with increasing STED power, from the confocal $F_1$ to the maximum STED $F_8$ (Fig. 3a). Foci outside the nucleus most likely report mitochondrial DNA and were therefore discarded from further analysis. Moreover, due to rRNA transcription and processing, as well as ribosome assembly, the nucleolus can be easily distinguishable from the remaining

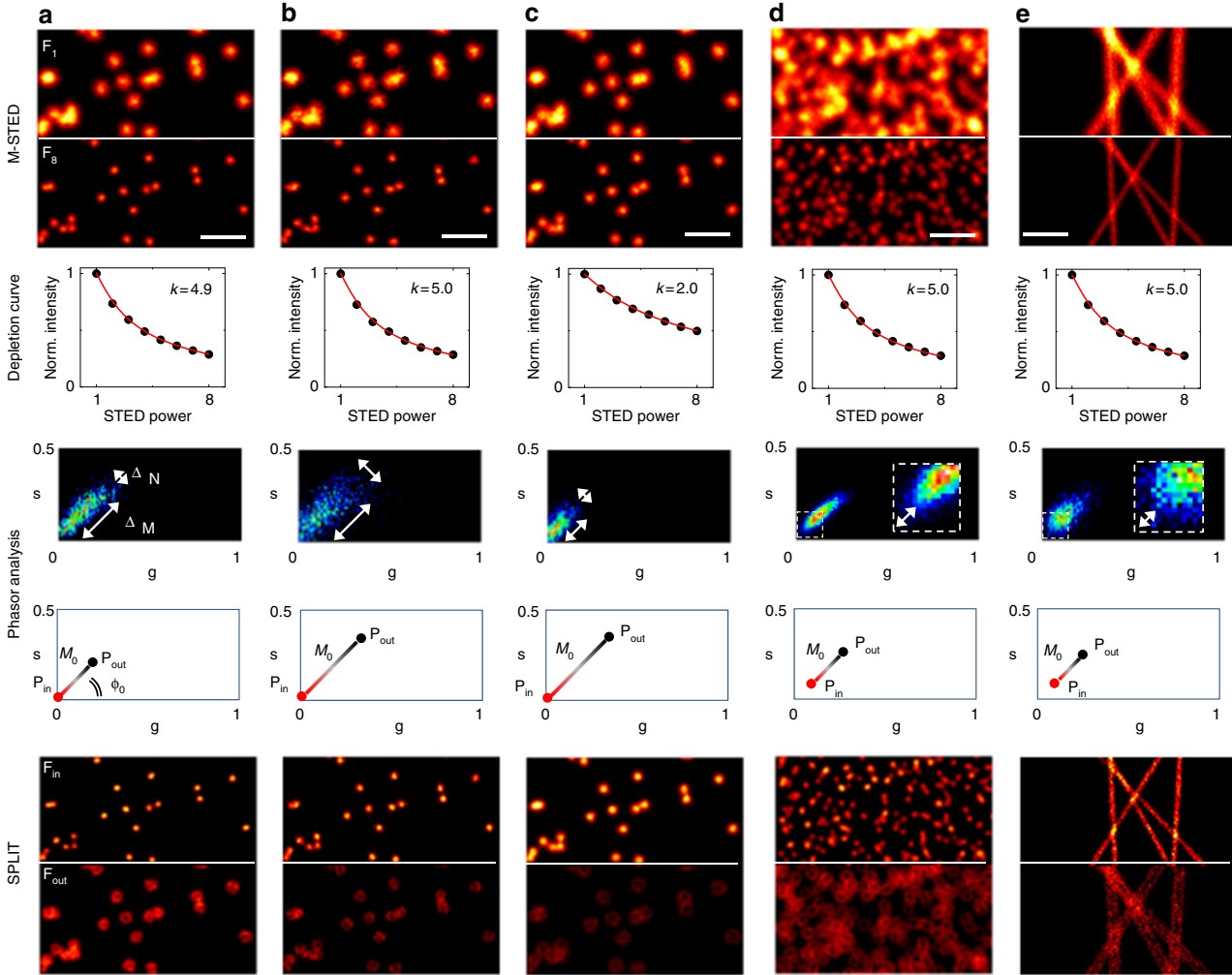

**Fig. 2** Depletion curve and phasor analysis in M-STED microscopy. **a–c** Simulations of sparse foci under different conditions of acquisition. Compared with **a**, **b** was simulated with the same maximum saturation value ($I_{max}/I_{sat} = 5$) but lower SNR ($S_{max} = 10$), while **c** had lower maximum saturation value ($I_{max}/I_{sat} = 2$) and the same SNR. **d** Simulation of crowded foci under the same conditions of acquisition of **a**. **e** Simulation of filaments under the same conditions of acquisition of **a**. Shown are, for all the simulations, from top to bottom: first and last images of the stack ($F_1$ and $F_8$), the average variation of the fluorescence intensity as a function of the STED power (depletion curve) and the estimated $k$ parameter (Eq. (4)), the corresponding phasor plots and the estimated positions of $P_{in}$ and $P_{out}$ vectors, and the final SPLIT image components $F_{in}$ and $F_{out}$. Scale bars: 1 µm

nucleus by a much higher degree of staining. These regions were thus also absent in the subsequent analysis.

As previously described, the fluorescence intensity variations along the stack were visualized into the phasor space (Fig. 3b), from which the modulation image $M(x,y)$ was created (Fig. 3c). The information contained in $M(x,y)$, together with $F_8$, was then used to apply the SPLIT and consequently to obtain a SPLIT image $F_{in}$ with enhanced resolution (Fig. 3d). $F_{out}$ contains the remaining, highly modulated photons originated away from the center of the foci that are excluded from the final image $F_{in}$. Importantly, $F_{out}$ also seems to contain structured features not directly related to the periphery ring of individual foci. This additional signal can therefore be associated to autofluorescence or out-of-focus foci. In either case, the fact that M-STED is able to separate this contribution from the final SPLIT image strongly supports its application to intact cells.

Imaging transcription foci using M-STED resulted in an improvement in resolution such that 2 foci that were initially barely undistinguishable in $F_8$ become completely resolved, due to the ability to efficiently separate distinct modulation contributions (Fig. 3a–d). As expected, the center of these nuclear foci coincides perfectly with the modulation minima as a consequence

of the modulation-based SPLIT. By fitting the line profiles in Fig. 3e with a multi-peak Gaussian function (not shown), we estimate the FWHM of these particular transcription foci to be ~181 and ~178 nm in $F_8$ ($P_{STED} = P_{max}$) and ~112 and ~90 nm in the SPLIT image. These values represent an improvement in resolution of roughly 80 nm, reinforcing the usefulness of M-STED to image biological samples even when compared to conventional STED microscopy.

Another quantitative feature of the M-STED method is the knowledge of the improvement of resolution introduced by the M-STED PSF with respect to the confocal PSF. As explained in the previous section, this information is extracted from the average depletion curve through the parameter $k$. In order to take advantage of this feature, we analyzed multiple images of transcription foci characterized by PSF of different sizes relative to confocal (Fig. 3f–j). These images were generated starting from M-STED stacks acquired at different values of $P_{max}$ (Fig. 3f, g, Supplementary Fig. 7a, b) and varying the processing parameters (see Methods) in order to obtain images of resolved foci with a good SNR (Fig. 3h, Supplementary Fig. 7c). For each image, the size of the PSF relative to confocal ($w/w_c$) was evaluated based on the value of $k$ and the processing parameters, resulting in values

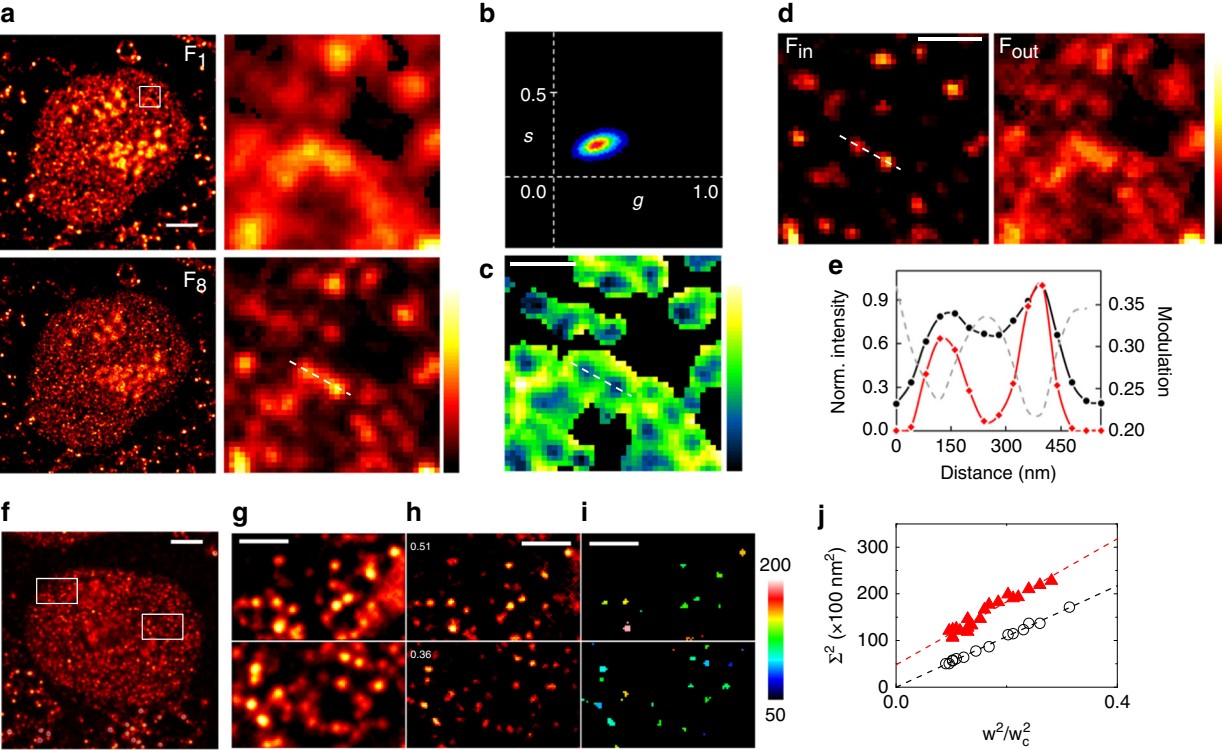

**Fig. 3** M-STED of transcription foci within intact MCF10A nuclei. **a** Example of acquired stack images, from the confocal ($F_1$) to maximum STED power ($F_8$) ($P_{max} = 43.3$ mW). Color scale: normalized intensity. Scale bar: 3 µm. **b** Phasor plot obtained from the data and **c** the corresponding modulation image $M(x,y)$. Color scale: modulation $M$ (0.16–0.51). Scale bar: 500 nm. **d** $F_{in}$ and $F_{out}$ images obtained by application of SPLIT to $F_8$. Color scale: normalized intensity. Scale bar: 500 nm. **e** Comparison of the ability to separate two foci within intact nuclei between $F_8$ (black circles) and $F_{in}$ (red diamonds), and the relation with the information encoded in $M(x,y)$. **f**–**j** Quantification of the size of the transcription foci. **f**, **g** Example of data used for size estimation acquired at $P_{max} = 24.2$ mW. Shown are the sum of the full-size M-STED stack (**f**) and two ROIs selected for the analysis (**g**). Scale bars: 3 µm in **f**; 1 µm in **g**. **h** $F_{in}$ images obtained from application of SPLIT to the ROIs, depicting resolved foci. Numbers indicate the estimated effective PSF size relative to the confocal ($w/w_c$). Scale bar: 1 µm. **i** Apparent size of individual foci shown in **h**. Scale bar: 1 µm. **j** Average $\Sigma^2$ values (Eq. (5)) extracted from each image such as represented in **i** (red) and from simulated images of point-like structures (black) shown in Supplementary Fig. 8. From the intercept (red), the size of the transcription foci was estimated to be 69 ± 5 nm

of $w/w_c$ that spanned roughly from 0.3 to 0.5 (Fig. 3h). From these multi-PSF images, it is possible to extrapolate the average size of the foci without any prior calibration of the microscope PSF. In fact, the apparent size $\Sigma$ of the foci in any of the images, resulting from the convolution of foci of size $\sigma$ with a PSF of waist $w$, and thus a full-width at half-maximum FWHM $= (2\ln2)^{1/2}w$, is given by: $\Sigma^2 = \sigma^2 + 2\ln2 w^2$. This can be written as

$$\Sigma^2 = \sigma^2 + (2\ln 2)w_c^2(w/w_c)^2. \tag{5}$$

The average value of $\Sigma^2$ extracted from each image (Fig. 3i) is reported in Fig. 3j as a function of the corresponding value of $(w/w_c)^2$. From the positive intercept we found that the average size of transcription foci is $\sigma = 69\pm5$ nm, in keeping with previous reports[33,34]. Interestingly, from the slope we found also a value for the waist of the confocal PSF of the microscope, $w_c = 221\pm6$ nm. Notably, the values of $\sigma$ and $w_c$ are obtained using values extracted from the images of foci, without any calibration of the microscope PSF. For comparison, the same analysis applied on simulated images of point-like structures (Supplementary Fig. 8) yielded a zero intercept and a value $w_c = 198\pm2$ nm (Fig. 3j).

**M-STED microscopy in the presence of STED-induced background**. When the absorption spectrum of a fluorophore is close to the STED wavelength, the STED laser may excite directly the fluorophores, adding a background to the STED image that is

generally visible as a doughnut-shaped halo[35]. This background can be efficiently removed from the fluorescence signal applying the SPLIT method to CW-STED and exploiting its different temporal fingerprint on the nanosecond timescale[28]. Instead, in our original M-STED design, the increase in STED power generates a linearly increasing background along the stack (Fig. 4a) whose phasor position, $\mathbf{P}_{bkgd}$, is co-aligned with the phasors $\mathbf{P}_{in}$ and $\mathbf{P}_{out}$, corresponding to the in and out regions of the PSF (Fig. 4b). In this configuration, the SPLIT algorithm cannot separate the background from the in-originated photons, and the expected improvement in resolution is lost (Supplementary Fig. 9). According to simulated M-STED data, the negative effect of the STED-induced background is significant when this background level is above 10% (Supplementary Fig. 9).

An efficient separation of background can be obtained by modulating both STED and excitation powers. We modified the depletion dynamics corresponding to the center of the PSF by modulating the excitation intensity along the stack as an exponential decay function, $I_{exc} \propto \exp(-t/\tau_{exc})$ (Supplementary Fig. 10). Consequently, the photons arising from the center of the PSF now present a specific decay along the stack that depends on the decrease of the excitation power (Fig. 4c). In the phasor plot, the "in" photons are no longer at the phasor position (0,0), being now present at position $\mathbf{P}'_{in}$ (Fig. 4d). In the periphery, fluorescence emission is influenced from both excitation and STED power variations, and thus the phasor shifts to the position $\mathbf{P}'_{out}$. Most importantly, since the STED-induced background

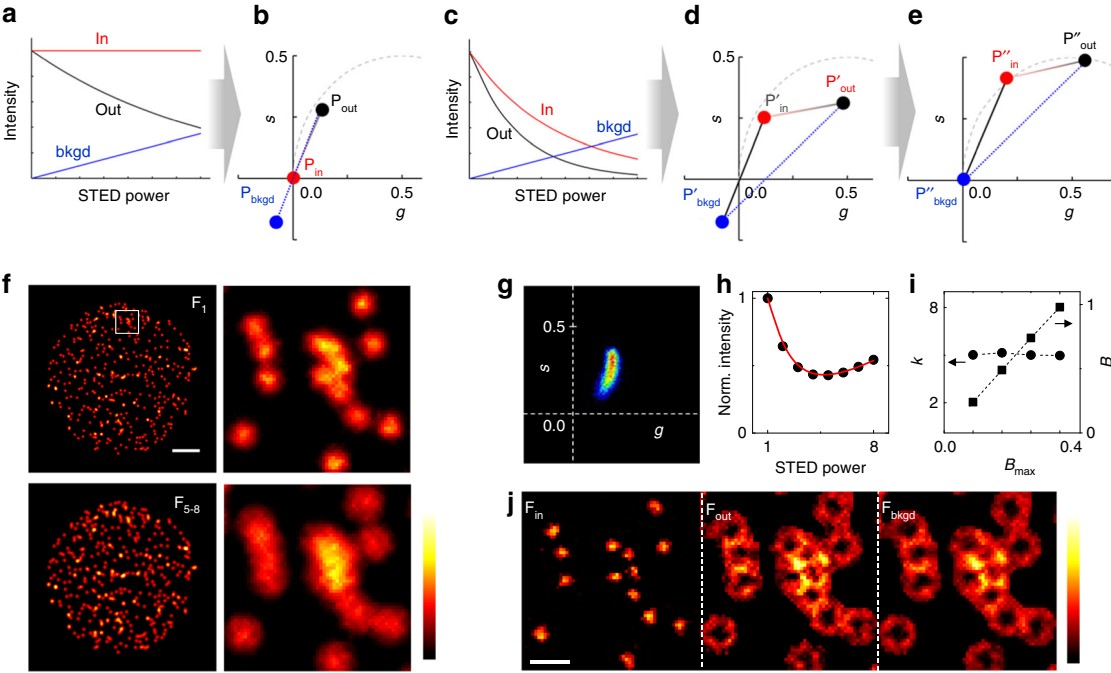

**Fig. 4** M-STED in presence of STED-induced background. **a** Fluorescence depletion dynamics in the center (in) and in the periphery (out) of the PSF, together with STED-induced background (bkgd) along the stack, for the original M-STED configuration. **b** Phasor representation of dynamics depicted in **a**. $P_{bkgd}$ is co-aligned with the phasors $P_{in}$ and $P_{out}$. **c** Fluorescence depletion dynamics resulting from the application of the modified version of M-STED, with modulation of both excitation and STED power as shown in the Supplementary Fig. 10. **d** Phasor representation of dynamics displayed in **c**. The photons originated in the center and the periphery of the PSF appear at new positions $P'_{in}$ and $P'_{out}$, respectively, and thus the three phasor $P'_{in}$, $P'_{out}$, and $P'_{bkgd}$ are no longer co-aligned. **e** For SPLIT application purposes, the coordinates are changed such that the position of the background phasor moved to the origin position. **f** Simulated imaging of nuclear foci with the modified version of M-STED, where the excitation intensity is modulated as an exponential decay (see Supplementary Fig. 10). Color scale: normalized intensity. Scale bar: 3 µm. **g** Phasor representation of dynamics encoded in **f**. **h** Average variation of the fluorescence intensity as a function of the STED power. The red line is a fit of Eq. (6) to the data. **i** Values of $k$ and $B$ retrieved from fitting the depletion curves of M-STED stacks simulated at different levels of background (Supplementary Fig. 11). **j** SPLIT components $F_{in}$, $F_{out}$, and $F_{bkgd}$, showing the efficient separation of $P'_{in}$, $P'_{out}$, and $P'_{bkgd}$. Color scale: normalized intensity. Scale bar: 500 nm

depends only on the STED power variations, the background photons appear in the same phasor position $P'_{bkgd} = P_{bkgd}$. The three components are therefore no longer co-aligned and can be efficiently separated with the SPLIT algorithm after a simple change of coordinates that moves the position of the background phasor to the origin (Fig. 4e, see Methods).

We simulated STED image stacks of nuclear foci using this modified version of M-STED (Fig. 4f–j). In this simulation, the final frames of the stack, which should contain better resolved foci, are smeared by the STED-induced background that counteracts the effect of the STED modulation (Fig. 4f). In the phasor plot, the presence of STED-induced background is revealed by a phasor bending towards the origin (Fig. 4g). Note that the different shape of the phasor (when compared with Fig. 1f) results from the fact that, in this case, the fluorescence arising from each pixel is represented in the phasor space as a combination of three different phasor positions ($P''_{in}$, $P''_{out}$, and $P''_{bkgd}$) as depicted in Fig. 4e.

In this configuration, both the modulation and phase are required to fully describe the position of the phasor. Moreover, since we need to separate three components ($P''_{in}$, $P''_{out}$, and $P''_{bkgd}$), we cannot use the simplified analysis shown in Fig. 2. Instead, we apply the SPLIT algorithm after assigning the positions of the phasors $P'_{in}$ and $P'_{out}$. These positions are calculated from a theoretical distribution of depletion decays described by $\gamma(r) = \gamma_0 + (I_{max}/I_{sat})(r^2/w_c^2)/T$, where $\gamma_0 = 1/\tau_{exc}$. The value of $I_{max}/I_{sat}$ is recovered by considering that in the new configuration the average depletion curve can be described by

(Supplementary Note 1)

$$\frac{\langle F(x,y,j)\rangle}{\langle F(x,y,1)\rangle} = e^{-\gamma_0(j-1)}\frac{1}{1+\frac{k}{2}\frac{(j-1)}{(n-1)}} + B\frac{(j-1)}{(n-1)}, \qquad (6)$$

where the brackets denote averaging over the entire image and $B$ is a constant proportional to the level of STED-induced background. Here, $k = I_{max}/I_{sat}$ represents again the saturation value at position $r = w_c$ for the image of the stack acquired at maximum STED power. When simulating M-STED images in the presence of different levels of background, the value of $k$ extracted from the fit was always in keeping with the simulated value of $I_{max}/I_{sat}$ (Fig. 4h, i, Supplementary Fig. 11). Application of SPLIT thus yields an $F_{in}$ image of higher resolution and simultaneously removes the STED-induced background (Fig. 4j).

We applied this configuration of M-STED to the direct visualization of replication foci labeled with Atto532 (Fig. 5). The final frames of the stacks, which should contain better resolved foci, are degraded by direct excitation of the Atto532 fluorophore by the 592-nm STED beam (Fig. 5a, e). The level of STED-induced background on a given experiment can be quantified from the average depletion curve (Fig. 5b–f) and visually evaluated on the phasor plot (Fig. 5c–g). The SPLIT algorithm successfully separates the super-resolved image $F_{in}$ from the images $F_{out}$ and $F_{bkgd}$ representing the fluorescence contributions from the periphery of the PSF and the STED-induced background, respectively (Fig. 5d–h). The improvement in resolution visible in the SPLIT image would not be possible

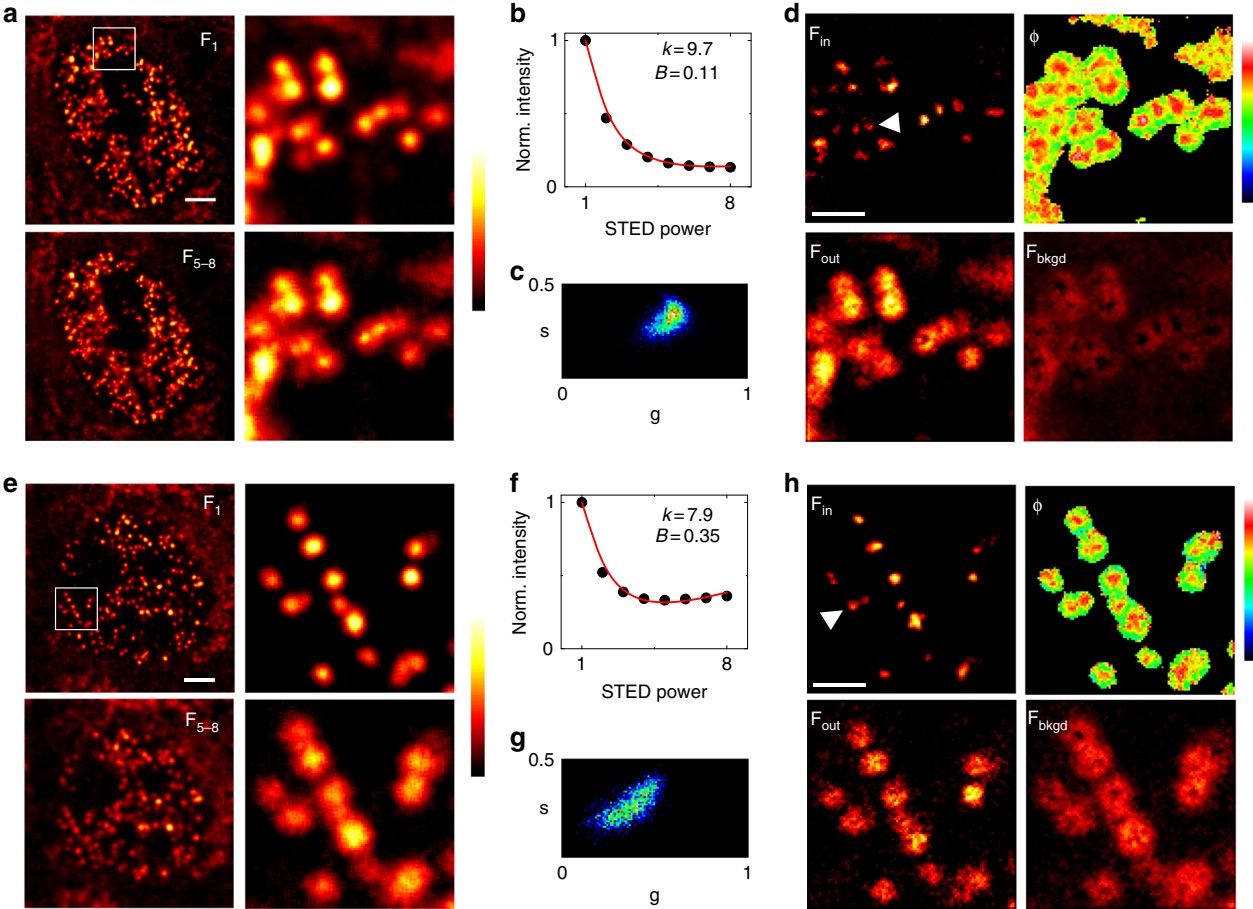

**Fig. 5** M-STED of replication foci in MCF10A nuclei. **a**, **e** Examples of acquired stack images, from the confocal $F_1$ to maximum STED power $F_8$ ($P_{max} = 19$ mW). Depicted is the sum of the last four frames $F_{5-8}$. Color scale: normalized intensity. Scale bars: 3 µm. **b**, **f** Average variation of the fluorescence intensity as a function of the STED power, together with the respective $k$ and $B$ values retrieved from the fitting of Eq. (6) to the data. **c**, **g** Phasor plot of the data shown in **a** and **e**, respectively. **d**, **h** SPLIT image components $F_{in}$, $F_{out}$, and $F_{bkgd}$, and the corresponding phase image $\phi$. The white arrowheads point at pairs of foci that are better resolved in the SPLIT images. Color scale: phase $\phi$ (0–0.75). Scale bars: 1 µm

without the removal of STED-induced background. The amount of background on a given pixel is essentially encoded in the modulation image $M(x,y)$. On the other hand, the phase image $\phi(x,y)$ encodes the spatial information required to separate photons stemming from the center and the periphery of the PSF (Fig. 5d–h).

It is worth noting that, in some cases, the level of STED-induced background can be minimized by proper settings of the instrumental conditions. For instance, in the example of Fig. 5a–d, where the measured background level is $B = 0.11$, one could have acquired the images at the same STED power but at a 10× higher excitation intensity. This would have produced a 10× smaller background level, $B/10 = 0.01$, which might be acceptable. Unfortunately, increasing the excitation power a given level is not always possible due to instrumental or sample limitations (e.g., increase in photobleaching), greatly justifying the use of M-STED. Importantly, this capability of M-STED to subtract the STED-induced background can be useful for the acquisition of multi-color STED images. This is especially relevant when a single STED wavelength is available and one of the fluorophores (the one whose absorption peak is closer to the STED wavelength) can be excited directly by the STED beam. An example of a SPLIT image of replication and transcription foci obtained by dual-color M-STED using a single STED wavelength is presented in Supplementary Fig. 12.

## Discussion

The spatial resolution that can be achieved in STED microscopy depends essentially on the level of saturation that can be reached by imaging a given sample. In many practical cases, limitations on the STED power used on a given experiment are imposed by detrimental effects that are more prominent at increasing average illumination powers. Examples of such unwanted effects are sample photobleaching, phototoxicity, or direct excitation by the STED beam. Here we have shown that modulating the STED power during the acquisition of a STED image may offer several technical advantages.

First, we have demonstrated that a simple analysis of the average depletion curve provides a quantitative value of the saturation level reached on a given STED experiment. Knowledge of the saturation value is important as it provides means to quantify the effective improvement of spatial resolution attained on a given STED experiment. Second, we have demonstrated that a modulation of the depletion power induces spatially dependent fluorescence intensity variations that can be analyzed to further enhance the spatial resolution of the STED image. The analysis of these variations (i.e., the fluorescence depletion dynamics) can be performed with the same SPLIT algorithm that was applied to time-resolved images acquired on a CW-STED microscope[28]. However, in M-STED the observed fluorescence depletion dynamics is not on the nanosecond timescale but rather depends

on the temporal scale of the modulation of the STED power. Thus, this method allows the application of the SPLIT algorithm in any STED microscope, without the need for pulsed excitation or dedicated lifetime detection hardware. In fact, STED images at different depletion powers can be acquired on any STED microscope. The modulation of the STED power can be obtained with any device capable of changing the beam intensity with a desired temporal pattern, synchronized with the image acquisition. For instance, in our experiments we have synchronized the modulation with the line scan repetition to avoid sample drift occurring between frames. In principle, one could generate an even faster modulation (e.g., within the pixel dwell time) using Acousto-Optical Modulators (AOM). In this respect, it is worth noting that the specific linear ramp pattern used in our experimental implementation (Supplementary Fig. 1a) is convenient to generate the specific form of the depletion curve described in Eq. (4) but is not necessary for the generation of a modulation image or a phasor plot. Any other pattern of STED modulation power, for instance a sinusoidal modulation at frequency $\omega$, synchronized with the image acquisition will produce a modulation of the fluorescence signal that can be used for SPLIT (Supplementary Fig. 1b). A simple M-STED acquisition with only two different STED powers can be sufficient to improve the resolution of the STED microscope (Supplementary Fig. 13), whereas a minimum of three different STED powers is required for the graphical analysis of the phasor plot (Supplementary Figs. 6 and 14). These could be all factors to take into account in more challenging applications such as 3D or live cell imaging (Supplementary Fig. 14), where minimization of photobleaching is critical. Nevertheless, we note that, at least in our experimental conditions, the use of a modulated STED power ramp per se does not seem to produce more photobleaching than a non-modulated STED acquisition delivering the same STED energy dose to the sample (Supplementary Fig. 15). Finally, we have also demonstrated that the unwanted background originating from direct excitation by a modulated STED beam can be removed by modulating also the power of the excitation beam. Conceptually, this is similar to a method that used modulated excitation light and frequency-dependent detection to unmix the fluorescence signal originated by the excitation beam from the background caused by the STED beam[36]. However, in ref.[36] the STED power was kept constant whereas in our approach we also exploit the potential advantages of a modulated STED beam.

As an application, we explored if and how modulation of the STED power could enhance the super-resolved STED imaging of nuclear transcription and replication foci. First of all, we have demonstrated that a modulation of the STED power from 0 to a value $P_{max}$ could be used to generate a SPLIT image of transcription foci with a better spatial resolution than in the corresponding STED image. We found that the effective spatial resolution attainable in the SPLIT image depends not only on the value of $P_{max}$ but also on the SNR of the M-STED stack. In this respect, in an effort to make the method more intuitive and accessible, we introduced a simplified SPLIT analysis that is easily performed within ImageJ. This simplified analysis is based on the graphical evaluation of the phasor spread in two orthogonal directions to estimate simultaneously the SNR of the image stack and the variations of dynamics induced by STED power modulation. The presence of a non-elongated phasor indicates that the encoded spatial information is hidden in the noise and cannot be used for SPLIT (Supplementary Fig. 5). This can be ascribed to non-optimized experimental conditions, such as a low STED power or a low SNR (Supplementary Fig. 5a), or to the fact that the sample does not contain diffraction-limited features (Supplementary Fig. 5b, c). In these cases, M-STED remains a useful quantitative tool but cannot be used to perform SPLIT.

Then, we exploited the self-calibration properties of the method to estimate the size of transcription foci. We demonstrated that knowledge of the STED saturation level, extracted from the average depletion curve, can be used to plot the apparent size of the foci as a function of the improvement in resolution and then generate a plot which is similar to those used in STED-based spot-variation FCS[30,31]. Notably, this spot-variation analysis was used to estimate the true size of both the foci and the confocal PSF directly from images of the sample, without prior calibration of the microscope PSF.

Finally, we exploited the subtraction of STED-induced background to perform imaging of replication foci labeled with a fluorophore whose absorption spectrum is closer the STED wavelength. The removal of STED-induced background can be especially useful for the acquisition of multi-color STED images. Enhanced STED imaging of basic biological processes such as transcription and replication opens interesting perspectives, providing a new tool complementary to genomic analysis and next-generation sequencing able to increase our comprehension of the fundamental mechanisms of basic cell biology.

The results of this work clearly demonstrate that the 'spectroscopy' approach to super-resolution microscopy proposed by SPLIT is not limited to the analysis of fluorescence lifetimes but can have a more general application in STED microscopy. This raises the question whether this approach could or not be applied also to other SRM techniques. In this respect, we note that our method is essentially based on the analysis of images obtained with a tunable PSF generated by varying the STED beam power (Fig. 1a). From this perspective, the measured intensity modulation (Fig. 1g) is a useful parameter as it is essentially a measure of the derivative of this variable PSF, as demonstrated in other contexts[37,38]. Thus, we expect that, in principle, a similar analysis could be applied to other SRM techniques based on the generation of images with multiple PSFs. In the near future, it will be interesting to test this idea with other super-resolution strategies, for instance those based on structured illumination[39] that are generally less phototoxic than STED-based methods and thus more compatible with the imaging of nuclear foci in live cells[14].

## Methods

**Cell culture and sample preparation.** MCF10A cells (ATCC CRL-10317[TM]) were grown in Dulbecco's modified Eagle's medium (DMEM) (Sigma Aldrich):Ham's F12K (Thermo Fisher Scientific) medium (1:1) containing 5% horse serum, 1% penicillin/streptomycin, 2 mM L-glutamine, 10 μg/mL insulin, 0.5 μg/mL hydrocortisone, 50 ng/mL cholera toxin (all from Sigma Aldrich), and 20 ng/mL EGF (PeproTech, Rocky Hill, NJ, USA) at 37 °C in 5% $CO_2$. Cells were grown on glass coverslips coated with 0.5% (w/v) gelatin (Sigma Aldrich) in phosphate-buffered saline (PBS). To detect replication and transcription, cells were incubated for 20 min with 10 μM 5-ethynyl-2′-deoxyuridine, EdU (Thermo Fisher Scientific), and 10 mM 5-bromouridine, BrU (Sigma Aldrich), respectively. Upon nucleotide incorporation, cells were fixed with 4% paraformaldehyde (w/v) for 10 min at room temperature.

After fixation, MCF10A cells were permeabilized with 0.1% (v/v) Triton X-100 in PBS for 10 min and incubated in blocking buffer (5% w/v bovine serum albumin (BSA) in PBS) for 30 min at room temperature.

For BrU detection, cells were incubated overnight at 4 °C with the primary antibody rabbit anti-BrdU (600-401-C29; Rockland Immunochemicals Inc., Limerick, PA, USA), in blocking buffer (1/500 dilution), followed by extensive washing. Cells were then incubated with the secondary antibody Chromeo488-conjugated anti-rabbit (ab60314; Abcam, Cambridge, UK) in PBS (1/100 dilution), for 45 min at room temperature, and washed with PBS.

EdU incorporation was detected using the Click-iT[TM] EdU imaging kit (Thermo Fisher Scientific) according to the manufacturer's instructions, but replacing the kit's azide by biotin azide to allow the subsequent immunostaining. After the click reaction, cells were washed with PBS and incubated with the primary antibody goat anti-biotin (600-101-098; Rockland Immunochemicals Inc.) in blocking buffer (1/100 dilution), for 1 h at room temperature. The secondary antibody Atto532-conjugated anti-goat (605-453-013; Rockland Immunochemicals Inc.) was then added to the cells for 1 h at room temperature (1/400 dilution), upon which cells were washed with PBS.

Before mounting, all coverslips were extensively washed with PBS and water. In case of double BrU/EdU staining, the protocols were followed sequentially, starting always with BrU immunostaining. In this case, before continuing to the EdU staining part, cells were fixed again for 5 min with 4% PFA and an additional blocking step was performed by incubating the cells for 30 min with blocking buffer loaded with IgG (015-000-002; Jackson ImmunoResearch Laboratories, Inc., West Grove, PA, USA).

HEK-293 cells and HeLa cells (ATCC) were grown in DMEM medium supplemented with 10% FBS, 1% penicillin–streptomycin, and 1% glutamine (all from Sigma Aldrich). For live cell imaging, HEK-293 cells were seeded on multi-well chambered cover glass (Ibidi, Munich, Germany) at 60–80% confluence. After 24 h, cells were transiently transfected with a plasmid encoding for EGFP-Δ50 lamin A[40] using Effectene reagent (Qiagen). The pEGFP-Δ50 lamin A was a gift from Tom Misteli (Addgene plasmid # 17653). For immunofluorescence imaging, HeLa cells were fixed with 3.2% PFA and 0.25% GA for 10 min at room temperature. After washing several times in PBS, cells were permeabilized using 0.3% Triton X-100 for 15 min and blocked for 30 min with 5% BSA. Next, the sample was incubated with primary antibody anti-α-Tubulin (T5168; Sigma Aldrich) in 5% BSA for 90 min at room temperature. After washing several times in PBS, cells were incubated with the secondary antibody AlexaFluor488-conjugated anti-mouse (A11001; Thermo Fisher Scientific) in 5% BSA at room temperature for 45 min.

Samples of 20-nm diameter yellow–green fluorescent spheres (Invitrogen) were prepared as follows. The spheres were diluted in water by 1:5000 (v/v) and the solution was then added to a poly-L-lysine (Sigma Aldrich)-coated glass coverslip. After 10 min the coverslip was washed with water, dried under a nitrogen flow, and finally mounted before the measurements.

**Experiments.** All imaging experiments were performed on a Leica TCS SP5 gated-STED microscope, using a HCX PL APO ×100 100/1.40/0.70 oil immersion objective lens (Leica Microsystems, Mannheim, Germany). Emission depletion was always accomplished with a 592 nm STED laser.

Excitation was provided by a white laser at the desired wavelength for each sample. Chromeo488 was excited at 488 nm and its fluorescence emission detected at 500–560 nm, with 1–10 ns time gating. Atto532 excitation was performed at 532 nm and the emission collected between 540 and 580 nm, with time gating of 2.5–6 ns. For two-color images, excitation/emission wavelengths were adjusted to 470/480–530 nm and 545/550–585 nm for Chromeo488 and Atto532, respectively.

For M-STED, stacks of STED images at different STED powers were obtained using the line sequential acquisition mode (400–700 Hz). STED power was measured at the aperture of the objective, at the sample plane (Supplementary Fig. 16). For the images acquired in the presence of STED-induced background, the excitation power was modulated as an exponentially decaying function with time-constant $\tau_{exc} = n/2$, where $n$ is the number of images forming the stack.

**STED phasor analysis and calculation of SPLIT images.** For a given image stack $F_j(x,y)$, the images of the phasor variables $g(x,y)$ and $s(x,y)$ were calculated as[41]

$$g(x, y) = \sum_{j=1}^{n} F_j(x, y) \cos[2\pi(j-1)/n] / \sum_{j=1}^{n} F_j(x, y)$$
$$s(x, y) = \sum_{j=1}^{n} F_j(x, y) \sin[(2\pi(j-1)/n)] / \sum_{j=1}^{n} F_j(x, y) \quad (7)$$

where $n$ is the number of images forming the stack. The modulation image $M(x,y)$ was calculated as $M(x,y) = (g^2(x,y) + s^2(x,y))^{1/2}$. The phase image $\phi(x,y)$ was calculated as $\phi(x,y) = \tan^{-1}(s(x,y)/g(x,y))$.

For M-STED data with negligible or no STED-induced background, we performed a separation into two components. The elongation of the phasor along the radial direction was quantified by the parameter $\Delta_M$, whereas its spread along the angular coordinate was quantified by the parameter $\Delta_N$. The parameters $\Delta_M$ and $\Delta_N$ were determined as the standard deviation of the values of $g$ and $s$ after rotation of an angle $-\phi_0$, where $\phi_0$ is the angle formed by the direction of elongation of the phasor with respect to the $g$-axis. For each pixel, the fraction $f_{in}(x,y)$ of fluorescence intensity associated with the center of the PSF was estimated by expressing the experimental phasor $\mathbf{P} = (g,s)$ as a combination of a phasor $\mathbf{P}_{in} = (g_{in},s_{in})$, representing fluorescence dynamics in the center of the PSF, and a phasor $\mathbf{P}_{out} = (g_{out},s_{out})$, representing fluorescence dynamics in the periphery of the PSF. The fraction $f_{in}$ was calculated as $f_{in} = (\mathbf{P}_{out}-\mathbf{P})\cdot(\mathbf{P}_{out}-\mathbf{P}_{in})/|\mathbf{P}_{out}-\mathbf{P}_{in}|^2$. This value is proportional to the distance between the phasor $\mathbf{P}$ and the phasor $\mathbf{P}_{out}$ along the line connecting $\mathbf{P}_{in}$ and $\mathbf{P}_{out}$ (Supplementary Fig. 3a), as described in the framework of the phasor analysis of FRET[42]. The values of $g_{in}$ and $s_{in}$ were determined as $g_{in} = M_{in}\cdot\cos\phi_0$ and $s_{in} = M_{in}\cdot\sin\phi_0$ respectively, where the value of $M_{in}$ was determined from the modulation histogram. The values of $g_{out}$ and $s_{out}$ were determined as $g_{out} = M_{out}\cdot\cos\phi_0$ and $s_{out} = M_{out}\cdot\sin\phi_0$, respectively, with $M_{out} = M_{in} + M_0$, where $M_0$ is an estimation of the smaller difference in modulation that can be distinguished at the available SNR. The value of $M_0$ was set on the basis of the elongation of the phasor along the radial direction, quantified by the parameter $\Delta_M$, compared to its spread in the orthogonal direction, quantified by

the parameter $\Delta_N$:

$$M_0 = (m_0/2)\Delta_N / |\Delta_M - \Delta_N|. \quad (8)$$

The value of the constant $m_0$ was set to 1 unless specified otherwise. Finally, the SPLIT image component $F_{in}(x,y)$ (i.e., the SPLIT image) was obtained multiplying a given STED image $F'_{STED}(x,y)$ by the fraction $f_{in}(x,y)$:

$$F_{in}(x, y) = f_{in}(x, y)F'_{STED}(x, y) \quad (9)$$

whereas the SPLIT image component $F_{out}(x,y)$ was obtained as $F_{out}(x,y) = (1-f_{in}(x,y))F'_{STED}(x,y)$.

For M-STED data with non-negligible STED-induced background we performed a separation into three components. The fraction $f_{in}(x,y)$ of fluorescence intensity associated to the center of the PSF was estimated using a slightly modified version of the SPLIT algorithm[28]. In this version, the value of unperturbed lifetime $\tau_0$ is set to the value $\tau_{exc}$ and the value of $k_S$ is set to the value $k_S = (I_{max}/I_{sat})(\tau_{exc}/(n-1))$. In addition to that, the phasor fingerprint of STED-induced background ($\mathbf{P}'_{bkgd}$) is subtracted from both the reference and experimental phasors prior to decomposition.

The whole image analysis was implemented in MATLAB (The Mathworks, Natick, MA).

**Simulations and data analysis.** Simulated STED image stacks of nuclear foci were generated using MATLAB. The object consisted in a variable number of point-like emitters distributed randomly inside a circular area with a diameter of 16 μm. Each stack consisted of a number $n$ of 512 × 512 frames with a pixel size of 40 nm. The maximum total number of photons detected from a single pixel position in one frame of the stack was set to $S_{max} = 30$ unless specified otherwise. For each frame $j$ of the stack, the object was convolved with the theoretical PSF of a STED microscope described by Eq. (3) setting $t = j-1$ and $T = n-1$. The maximum value of the STED saturation level $I_{max}/I_{sat}$ was set as specified and the obtained image was corrupted by Poisson noise. The waist of the confocal PSF was always set to $w_c = 200$ nm, unless otherwise stated. The large foci (Supplementary Fig. 5b) were simulated substituting each point-like emitter with a 10 pixels-sized structure. The unresolved foci (Supplementary Fig. 5c) were simulated by increasing the density of foci and using a larger confocal PSF ($w_c = 400$ nm). In order to simulate the effect of STED-induced background, a doughnut-shaped excitation PSF was added to the model. The maximum number of STED-induced background photons detected from a single pixel position in one frame of the stack was set to a percentage $B_{max}$ of the maximum signal $S_{max} = 40$. In these simulations, the excitation power was either kept constant (data reported on Supplementary Fig. 9) or modulated as an exponentially decaying function with time-constant $\tau_{exc} = n/2$.

The analysis of size was performed on rectangular ROIs extracted from experimental M-STED stacks of transcription foci acquired at different values of $P_{max}$. For each ROI, a SPLIT image was obtained by setting the positions of the phasors $\mathbf{P}_{in}$ and $\mathbf{P}_{out}$ in order to achieve the maximum possible separation of foci in the image. The apparent size $\Sigma$ of the foci in the SPLIT image was quantified by means of an algorithm based on image correlation spectroscopy that maps the size of the structures in an image[43]. For each SPLIT image, the relative size of the effective PSF with respect to confocal, $w/w_c$, was estimated starting from the value of the parameter $k$, obtained from a fit of the average depletion curve, and the positions of the phasors $\mathbf{P}_{in}$ and $\mathbf{P}_{out}$ used for processing the image.

**Code availability.** The custom codes are available from the corresponding author upon reasonable request.

**Data availability.** The data that support the findings of this study are available from the corresponding author upon reasonable request.

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

## Acknowledgements

This work was supported in part by Fondazione Cariplo and Associazione Italiana per la Ricerca sul Cancro (AIRC) through Trideo (Transforming Ideas in Oncological Research) Grant ID 17215.

## Author contributions

L.L., M.J.S., and A.D. designed research. M.J.S., L.F., M.F., S.P., and L.P. prepared samples. M.J.S., L.L., and P.B. performed experiments. L.L., M.O., and L.S. wrote software. All authors analyzed and discussed results. M.J.S. and L.L. wrote the manuscript with input from A.D., G.I.D., P.G.P. and all other authors.

## Additional information

**Competing interests:** The authors declare no competing interests.

