## [Peer Review File · Nature Communications]

Reviewers' comments:

Reviewer #1 (Remarks to the Author):

Lanzano and co-workers present an elegant way to further optimize the performance of super-resolution STED microscopy. They achieve this by taking advantage of the tunability of the STED microscope caused by changing the intensity of the added STED laser. Fantastic approach and FINALLY another one using this unique feature of the STED microscope (previously only employed in a different way in the STED-based spot-variation FCS approach)! The authors give compelling arguments, controls and experiments. This is an important story that on the first sight might be very specific and relevant to only a few experts. Yet, I think it is worth publishing in Nature Communications to a broader audience. However, the authors need to tune their manuscript more towards this general readership.

The manuscript does not contain any flaws and the line of argumentation/results is very logic. However, the presentation contains too many special wordings and expects too much special knowledge by the reader – but this can be easily corrected by some re-writing.

- It is tough to extract the most essential part, the usage of the unique tunability of the STED microscope. Rather than using the wording "dynamics" (which by the way is completely confusing since the reader expects measurements of cellular dynamics or similar) bring this tunability (also already in the title). Maybe introduce the power of this capability by already bringing sv-FCS in the introduction (and not just in the discussion).
- Make a clearer distinguishing from the lifetime-based SPLIT approach. One gets confused from the introduction of SPLIT, since it is associated with the lifetime-approach from reference 28. Clearly write that SPLIT is rather a way of extracting components of different spatial resolution via phasor-plot information – I got several times confused between the new and old (ref 28 + necessity of lifetime-electronics and CW-STED) approach.
- Some of the expressions (g, s, Pin, Pout, DL-PSF etc) are introduced once and then used way later – sometimes it is good to give a reminder.
- Some of the expressions such as STED decay sound very confusing and not very specific (what is a STED decay or a SPLIT image or a fluorescence emission decay, could be many things?) – Please check whether this can be defined better.

Minor comments:

- Line 132: Bring one sentence on the way the simulations were done.
- Line 146: Why significant?
- Lines 149ff: Did not get the argumentation here?
- Equation 3: Give a better reasoning for this equation (maybe in the SI).
- Line 167: This calculation is not immediately obvious.
- Lines 211/212: Did not understand this argumentation.
- Line 277: Why arbitrary?
- Line 281: What is meant by artificial?
- Lines 330ff: The multi-color discussion needs more explanation – why is it especially important here?
- Lines 368ff: Hard to understand the argumentation here – what is meant by "any modulation"? The modulation somehow needs to be correlated to the acquisition protocol (such as a line scanning frequency or similar).

Reviewer #2 (Remarks to the Author):

This is a sound, innovative study of general interest. The paper is a generalization of this group's previous SPLIT approach published in the same journal in 2015. This time the dynamics are encoded within STED images acquired at different STED laser powers instead of the fluorophore lifetime (as in ref 28).

Pros: This new method can be readily used on any STED microscope-as a matter of fact they used a commercial Leica SP5 STED. It is self-calibrating and could potentially be extended to any SR methods in which one can re-engineer the PSF. This is an intriguing and potentially impactful aspect.

Open questions: Since it takes about 8 images for each focal plane, is it suitable for 3D-STED or would that be prohibited due to increase in Photobleaching?

By the same token, can this be applied to live STED (3D/2D)?

What's the post-acquisition data processing time like?

It seems like the transcription foci which are measured at 69nm could be imaged within the specs of the Leica STED without the modulation. Therefore, the investigators should provide data on structures in the 20nm range (beads?) to demonstrate the added value of the technique.

Only the 592 nm depletion line was tested. Can that be generalized?

Reviewer #3 (Remarks to the Author):

This paper presents a very elegant out-of-the-box generalisation of a method that has been published by the same authors with immediate applicability and utility for current STED users. It provides higher resolution and background suppression at very little experimental and zero instrumental cost. The method is based on the realisation that fluorescence emission outside the perfect zero center of the TED doughnut, i.e. overlapping with the gradient STED intensity of the saturated doughnut, experiences a spatially-encoded quenching that can be decoded spectroscopically. This way, these photons can be made to contribute to the super-resolved image.

In a previous paper by the same authors, the fundament of the method was laid out: differential quenching results in fluorescence lifetime changes. This additional information can be conveniently graphically analysed by its polar representation in "phasor space" to unmix the perfectly unperturbed photon signals that give rise to the super-resolved image. This method, however, demands lifetime detection capabilities in the microscope which is not routinely available and comes with its own set of technical challenges.

At the heart of the method is the phasor analysis of fluorescence dynamics, i.e. time profiles. In this paper, a clever alternative for the nanosecond lifetime dynamics is presented: rather than relying on the intrinsic dynamics of fluorescence, the additional dynamics information that is used to extract higher resolutions is introduced extrinsically by the modification of the STED intensities, again producing spatially encoded variations in the fluorescence emission that can be regarded to occur "in time" so that they can be unmixed by the phasor method. In a way, this amounts to the exploitation of a clever lock-in detection scheme.

The paper clearly explains the relation to and logical progression from the use of lifetime to the implementation of intensity modulation. Using simulation and real biological application, the method is convincingly demonstrated, and the effects of SNR and crowding that limit the method are investigated. The unique self-calibration property of the method is used for the quantification of PSF size, and thus physical size of objects. Furthermore, using a modification of the approach simulating fluorescence decay, it is demonstrated that the undesired excitation of the fluorophore by the STED laser can be unmixed.

In conclusion, this manuscript completes a general approach introduced by the same authors with

an extremely useful and experimentally important implementation. The paper thus provides a significant intellectual and practical advance to the field of STED and other super-resolution microscopies.

I have a few minor points:

- Although we need the concept of the "arbitrary timescale T" (line 114-115) in order to align the idea with the use of phasor analysis, the point should be made that the timing of the acquisitions is irrelevant. Time here refers to the rank order of the images in a stack and it would be helpful if this can be expressed as such, maybe by referring to a T'.

- How sensitive is the method to the occurrence of bleaching in the individual acquisitions? Can this be counter-acted by a pseudo random order of the STED intensities, or by a forward-reverse approach, or does progressive bleaching cancel out (/blend in) as a constant?

- How many images with different STED powers would be sufficient for a resolution increase, i.e. how does resolution improvement depend on the number of STED powers used? This should be easy to simulate and would provide adopters of the method with a guideline for their experiments.

- I realize that the resolution improvement that can be reached depends on variables, but it would be nice to have a factor increase in resolution stated for the simulation and experimental examples.

- To better understand the necessity for the Σ^2 determination, I would like the authors to discuss the origin of variability in the effective PSF (expressed as w/w_c).

- Am I correct to assume that the gap in the phasor in crowded images is caused by spill-over of "out" fluorescence to any "in" location?

- I think the authors are missing an important opportunity to support their statement that one of the advantages of M-STED is the suppression of autofluorescence, i.e. fluorescence not generated by the specific fluorophore. At the approximate location of line 231, I would point out that F_{out} in Fig 3d not only shows the typical "defocus ring" around the point-like structures as is seen in the simulations, but also seemingly structured information in different image locations. I assume much of this signal is autofluorescence.

In the same vein, "background" in e.g. line 411 should read "STED excitation background".

- The sentences "In order to test this idea, we simulated STED As expected, ... are indeed smeared" (Lines 291-293) are weirdly put. Observations in simulations confirm the validity of the model, and should not be discussed as if observing real data. This is what Fig 5 is for. "As expected" should be "Consequently" and "indeed" should be removed.

- The "bending towards the origin" (Line 294) is not sufficiently explained as it is not congruent with the depiction in Fig 4d.

- The removal of STED-induced dye excitation, resulting in background emission requires the additional modulation of the excitation power. The adopter of the method should have a means to judge if this difficulty is big enough to justify the increase increase in experimental complexity. Fig 5 demonstrates a case in which this is necessary, but it would be helpful if the simulation in Fig 4 can be expanded (Suppl. Information) to include the "conventional" M-STED analysis, i.e. ignoring the presence of background. At what levels of background is a switch to the more complicated method justified?

I understand that, in the end, hopefully, producers of STED equipment will implement M-STED in its two implementations as this is only a matter of software control over the settings during an acquisition series, so that the "experimental complexity" reduces to the press of a button, but a

guideline toward yes-or-no worrying about STED excitation in nevertheless warranted.

- The authors should consider including in the main body of the manuscript, the explanation on the unmixing of "in" and "out" information in phasor space (the SPLIT method) as is explained in the methods section (Lines 637-640). This is, the position of points along the connecting line between the "in" and "out" positions represent their mixing ratios, allowing the extraction of the fraction of "in" intensity in the pixel cloud.

- The legends for the figures sometime repeat information given in the main body of the text and can be shortened.

Reviewers' comments:

We thank all the reviewers for the very useful comments. All major modifications are highlighted in red in the manuscript.

Reviewer #1 (Remarks to the Author):

Lanzano and co-workers present an elegant way to further optimize the performance of super-resolution STED microscopy. They achieve this by taking advantage of the tunability of the STED microscope caused by changing the intensity of the added STED laser. Fantastic approach and FINALLY another one using this unique feature of the STED microscope (previously only employed in a different way in the STED-based spot-variation FCS approach)! The authors give compelling arguments, controls and experiments. This is an important story that on the first sight might be very specific and relevant to only a few experts. Yet, I think it is worth publishing in Nature Communications to a broader audience. However, the authors need to tune their manuscript more towards this general readership.

We thank the reviewer for the very useful comments. We modified the manuscript following the reviewer's suggestions. Below we report our point-by-point reply.

The manuscript does not contain any flaws and the line of argumentation/results is very logic. However, the presentation contains too many special wordings and expects too much special knowledge by the reader – but this can be easily corrected by some re-writing.

- It is tough to extract the most essential part, the usage of the unique tunability of the STED microscope. Rather than using the wording “dynamics” (which by the way is completely confusing since the reader expects measurements of cellular dynamics or similar) bring this tunability(also already in the title). Maybe introduce the power of this capability by already bringing sv-FCS in the introduction (and not just in the discussion).

We thank the reviewer for these comments. We carefully rewrote several parts of the text taking into account the comments of the reviewer.

In particular:

- We modified the text (in the title, in the abstract and in the introduction) to highlight the STED ‘tunability’ concept
- We cited the example of spot variation FCS already in the introduction
- We introduced the new Eq.2 (STED spatial resolution as a function of the depletion power)
- We modified the text to avoid the unnecessary use of the wording ‘dynamics’ in the abstract and introduction.
- We introduce and justify the use of the term ‘dynamics’ in Results:

“In other words, the signal from fluorophores located in different parts of the PSF will show intensity variations in the temporal channel that are function of the fluorophore’s position within the PSF. In formal analogy to Ref. 28, we may refer to these variations as ‘fluorescence depletion dynamics’. However, while in Ref.28 the word dynamics was used to indicate the nanosecond fluorescence lifetime, here it denotes the intensity variations in response to a modulation of the STED power that can be implemented at an arbitrary timescale.”

- Make a clearer distinguishing from the lifetime-based SPLIT approach. One gets confused from the introduction of SPLIT, since it is associated with the lifetime-approach from reference 28. Clearly write that SPLIT is rather a way of extracting components of different spatial resolution via phasor-plot information – I got several times confused between the new and old (ref 28 + necessity of lifetime-electronics and CW-STED) approach.

We thank the reviewer for this comment. We modified the text following the suggestion of the referee.

-In the introduction:

“the phasor plot is used to decode spatial information encoded in a temporal channel and extract components of different spatial resolution. For instance, in the specific implementation described in Ref.28, a STED beam was used to generate a fluorescence lifetime gradient across the detection volume and the phasor plot was used to decode the spatial information encoded within this lifetime gradient.”

-We removed ambiguous expressions such as: “As in the original SPLIT”.

-In the Results:

“While in Ref.28 the intrinsic fluorescence decay of the fluorophore (measured in the nanosecond time scale) was used as means to separate *in* and *out* originated photons, here the same goal is achieved by analyzing STED images of variable resolution.”

- Some of the expressions (*g*, *s*, *P_{in}*, *P_{out}*, DL-PSF etc) are introduced once and then used way later – sometimes it is good to give a reminder.

Thanks for the suggestion. We carefully checked their definitions and added reminders of the definitions of these expressions throughout the text.

- Some of the expressions such as STED decay sound very confusing and not very specific (what is a

STED decay or a SPLIT image or a fluorescence emission decay, could be many things?) – Please check whether this can be defined better.

Thanks for the suggestion. In the revised version, we removed most of the confusing expressions. For instance, instead of the expression ‘STED decay’ we now use ‘depletion curve’, as defined in the text. We also clearly define what a SPLIT image is and what are SPLIT image components (F_{in} , F_{out}).

Minor comments:

- Line 132: Bring one sentence on the way the simulations were done.

Thanks for the suggestion. A brief explanation on how the simulation was performed was added to the manuscript as follows:

“To create such image stack, point-like objects were convolved with the theoretical PSF of a STED microscope described by Eq.1, with STED intensity I_w varying from 0 in the first frame to I_{max} in the last frame.”

- Line 146: Why significant?

We mean above the noise level.

- Lines 149ff: Did not get the argumentation here?

As described in the Methods, the SPLIT image is simply obtained by multiplying the STED image and the fraction $f_{in}(x,y)$. As a ‘STED image’, we can use any of the images of the stack or their sum. We rewrote these lines to avoid confusion:

“Note also that the SPLIT algorithm can be used to improve the spatial resolution of any of the STED images available in the stack. For instance, one can generate the SPLIT image using a sum of the images of the stack (Supplementary Fig.4)

- Equation 3: Give a better reasoning for this equation (maybe in the SI).

Thank you. We added Supplementary Note 1.

- Line 167: This calculation is not immediately obvious.

It is straightforward now that we introduced the new Eq.2.

- Lines 211/212: Did not understand this argumentation.

For better clarification, the following was added to the manuscript:

“In other words, in crowded samples, the modulation is different from zero even in the center of the foci (where in principle $I_{\text{STED}}=0$ and fluorescence is constant throughout the stack) because fluorescence coming from the periphery of other foci (and thus modulated) is also detected in every pixel.”

Then, when we choose the positions of \mathbf{P}_{in} and \mathbf{P}_{out} , we are indeed approximating a continuous distribution of decay rates with a simpler two-component model. For this reason, setting \mathbf{P}_{in} at a position corresponding to the minimum modulation observed in the data (different from 0 in crowded samples) provides a better approximation of the data.

- Line 277: Why arbitrary?

We apologize for the confusion. We rewrote the sentence:

“Consequently, in the center of the PSF the fluorophore emission decays from P_1 to P_8 (Figure 4c).”

- Line 281: What is meant by artificial?

We apologize for the confusion. We rewrote the sentence:

“The photons arising from the center of the PSF now present a specific decay along the stack...”

- Lines 330ff: The multi-color discussion needs more explanation – why is it especially important here?

In our setup, we have a single STED wavelength at 592nm and the minimum excitation wavelength available is 470nm. The most convenient combination of fluorophores for 2-color STED imaging was the one selected in this work (Chromo-488 and Atto-532). We wrote:

“Importantly, this capability of M-STED to subtract the STED-induced background can be useful for the acquisition of multi-color STED images. This is especially relevant when a single STED wavelength is available and one of the fluorophores (the one whose absorption peak is closer to the STED wavelength) can be excited directly by the STED beam.”

- Lines 368ff: Hard to understand the argumentation here – what is meant by “any modulation”? The modulation somehow needs to be correlated to the acquisition protocol (such as a line scanning frequency or similar).

We apologize for the confusion. Clearly, the modulation needs to be synchronized with the acquisition. We meant ‘any other pattern of modulation’. We modified the text to avoid confusion:

“Any other pattern of STED modulation power, for instance a sinusoidal modulation at frequency ω synchronized with the image acquisition, will produce a modulation of the fluorescence signal that can be used for SPLIT (Supplementary Fig.1b).”

Reviewer #2 (Remarks to the Author):

This is a sound, innovative study of general interest. The paper is a generalization of this group's previous SPLIT approach published in the same journal in 2015. This time the dynamics are encoded within STED images acquired at different STED laser powers instead of the fluorophore lifetime (as in ref 28).

Pros: This new method can be readily used on any STED microscope-as a matter of fact they used a commercial Leica SP5 STED. It is self-calibrating and could potentially be extended to any SR methods in which one can re-engineer the PSF. This is an intriguing and potentially impactful aspect.

We thank the reviewer for all the useful comments. We modified the manuscript following the reviewer's suggestions. Below we report our point-by-point reply.

Open questions: Since it takes about 8 images for each focal plane, is it suitable for 3D-STED or would that be prohibited due to increase in Photobleaching?

By the same token, can this be applied to live STED (3D/2D)?

We thank the referee for raising this important point. The method requires the collection of n images at different STED powers but in certain applications (e.g. 3D or live) it is important to minimize the acquisition time and thus photobleaching.

In this respect we note that we do not need 8 different values of STED power. In fact, an improvement of resolution can be obtained already with $n=2$ values of STED powers. With a minimum of $n=3$ values one can also perform the graphical analysis of the phasor plot. This is now explained in the text and new Supplementary Fig6. As an experimental proof of principle, we tested the $n=2$ acquisition with a sample of 20 nm beads (Supplementary Fig.13) and the $n=3$ acquisition with a live cell sample (Supplementary Fig.14).

We now discuss these points in the main text:

“A simple M-STED acquisition with only two different STED powers can be sufficient to improve the resolution of the STED microscope (Supplementary Fig.13), whereas a minimum of three different STED powers is required for the graphical analysis of the phasor plot (Supplementary Fig.6, Supplementary Fig.14). These could be all factors to take into account in more challenging applications such as 3D or live cell imaging (Supplementary Fig.14), where minimization of photobleaching is critical.”

What's the post-acquisition data processing time like?

In general, the processing time depends on the size of the stacks. In our most user-friendly Matlab script (which includes some file opening/parameters input dialogues), the total processing of a

120x120x8 stack takes less than about 20 sec while the processing of a 512x512x8 stack takes less than about 1 minute.

However, we strongly believe that the processing time can be further optimized.

It seems like the transcription foci which are measured at 69nm could be imaged within the specs of the Leica STED without the modulation. Therefore, the investigators should provide data on structures in the 20nm range (beads?) to demonstrate the added value of the technique.

We added Supplementary Fig.13 that shows data obtained with 20 nm beads. The apparent size of the beads in the SPLIT image is about 40 nm whereas in the STED image is about 60 nm.

Only the 592 nm depletion line was tested. Can that be generalized?

We tested only the 592 nm depletion line because this is the only one available in our commercial STED setup. In principle, since the method is based on the ability to tune/modulate the STED power, its application to other STED wavelengths should be straightforward. As long as the STED intensity can be modulated along an image stack, M-STED should be applicable. Nonetheless, for the same fluorophore, the k values and consequent improvement in resolution cannot probably be directly compared. However, this is mostly a matter of different depletion efficiency (and possible STED-induced background) that is not directly related to M-STED but rather a general feature of STED microscopy.

Reviewer #3 (Remarks to the Author):

This paper presents a very elegant out-of-the-box generalisation of a method that has been published by the same authors with immediate applicability and utility for current STED users. It provides higher resolution and background suppression at very little experimental and zero instrumental cost. The method is based on the realisation that fluorescence emission outside the perfect zero center of the TED doughnut, i.e. overlapping with the gradient STED intensity of the saturated doughnut, experiences a spatially-encoded quenching that can be decoded spectroscopically. This way, these photons can be made to contribute to the super-resolved image. In a previous paper by the same authors, the fundament of the method was laid out: differential quenching results in fluorescence lifetime changes. This additional information can be conveniently graphically analysed by its polar representation in "phasor space" to unmix the perfectly unperturbed photon signals that give rise to the super-resolved image. This method, however, demands lifetime detection capabilities in the microscope which is not routinely available and comes with its own set of technical challenges. At the heart of the method is the phasor analysis of fluorescence dynamics, i.e. time profiles. In this paper, a clever alternative for the nanosecond lifetime dynamics is presented: rather than relying on

the intrinsic dynamics of fluorescence, the additional dynamics information that is used to extract higher resolutions is introduced extrinsically by the modification of the STED intensities, again producing spatially encoded variations in the fluorescence emission that can be regarded to occur "in time" so that they can be unmixed by the phasor method. In a way, this amounts to the exploitation of a clever lock-in detection scheme.

The paper clearly explains the relation to and logical progression from the use of lifetime to the implementation of intensity modulation. Using simulation and real biological application, the method is convincingly demonstrated, and the effects of SNR and crowding that limit the method are investigated. The unique self-calibration property of the method is used for the quantification of PSF size, and thus physical size of objects. Furthermore, using a modification of the approach simulating fluorescence decay, it is demonstrated that the undesired excitation of the fluorophore by the STED laser can be unmixed.

In conclusion, this manuscript completes a general approach introduced by the same authors with an extremely useful and experimentally important implementation. The paper thus provides a significant intellectual and practical advance to the field of STED and other super-resolution microscopies.

We thank the reviewer for all the useful comments. We modified the manuscript following the reviewer's suggestions. Below we report our point-by-point reply.

I have a few minor points:

- Although we need the concept of the "arbitrary timescale T " (line 114-115) in order to align the idea with the use of phasor analysis, the point should be made that the timing of the acquisitions is irrelevant. Time here refers to the rank order of the images in a stack and it would be helpful if this can be expressed as such, maybe by referring to a T' .

In the revised version, we rewrote many parts of the text to clarify that what is important is to have images at different power and the timing of acquisition is irrelevant. For instance:

"In formal analogy to Ref. 28, we may refer to these variations as 'fluorescence depletion dynamics'. However, while in Ref.28 the word dynamics was used to indicate the nanosecond fluorescence lifetime, here it denotes the intensity variations in response to a modulation of the STED power that can be implemented at an arbitrary timescale."

"While in Ref. 28 the intrinsic fluorescence decay of the fluorophore (measured in the nanosecond time scale) was used as means to separate *in* and *out* originated photons, here the same goal is achieved by analyzing STED images of varying resolution."

To avoid confusion, we indicate the number of images in a stack with n instead of T .

- How sensitive is the method to the occurrence of bleaching in the individual acquisitions? Can this be counter-acted by a pseudo random order of the STED intensities, or by a forward-reverse approach, or does progressive bleaching cancel out (/blend in) as a constant?

We thank the reviewer for this comment. In all the experiments reported in the manuscript, the STED power was modulated during line repetition. At the conditions of our experiments, no significant bleaching was observed along the stack (i.e. it was occurring at a much slower rate compared to the line time).

However we totally agree with the referee that it will be interesting to explore in the future if and how changing the order of the STED powers could result in an improvement of the performances in the presence of photobleaching.

- How many images with different STED powers would be sufficient for a resolution increase, i.e. how does resolution improvement depend on the number of STED powers used? This should be easy to simulate and would provide adopters of the method with a guideline for their experiments.

We thank the reviewer for this comment. We added Supplementary Fig. 6 and citing text:

“It’s worth noting that the minimum number of frames required to calculate a modulation is $n=2$. However, for $n=2$ the value of the phasor coordinate s is systematically null (see Equation 7) and in this case we cannot take advantage of the graphical analysis of the phasor plot (Supplementary Fig.6a). The minimum number of frames required to perform a graphical analysis of the phasor plot is $n=3$ (Supplementary Fig.6b,c). The use of larger values of n allows the calculation of the phasor plot also at higher frequencies (Supplementary Fig.6d).”

- I realize that the resolution improvement that can be reached depends on variables, but it would be nice to have a factor increase in resolution stated for the simulation and experimental examples.

We thank the reviewer for this very important comment. From the line profiles shown in figures 1 and 3 (and in the new Supplementary figures 13 and 14), FWHM values were determined and added to the text to allow for a better evaluation of the improvement in resolution achieved through M-STED.

- To better understand the necessity for the Σ^2 determination, I would like the authors to discuss the origin of variability in the effective PSF (expressed as w/w_c).

For the data reported in Fig.1j, the different values of effective PSF (expressed as w/w_c) have been generated in two ways. First, by acquiring M-STED data at very different values of P_{\max} (ranging between about 5 mW and 43 mW). Second, by processing each M-STED acquisition with slightly different parameters (i.e. by setting different values of P_{in} and P_{out} and/or by using a sum of the images of the stack instead of only the last frame). For instance, using M-STED data acquired at P_{\max}

=43 mW, we obtained SPLIT images with values of w/w_c around 0.3 while using M-STED data acquired at $P_{\max}=24$ mW, we obtained SPLIT images with values of w/w_c between 0.35 and 0.5.

- Am I correct to assume that the gap in the phasor in crowded images is caused by spill-over of "out" fluorescence to any "in" location?

Indeed that is the case. For better clarification, the following was added to the manuscript:

"In other words, in crowded samples, the modulation is different from zero even in the center of the foci (where in principle $I_{\text{STED}}=0$ and fluorescence is constant throughout the stack) because fluorescence coming from the periphery of other foci (and thus modulated) is also detected in every pixel."

- I think the authors are missing an important opportunity to support their statement that one of the advantages of M-STED is the suppression of autofluorescence, i.e. fluorescence not generated by the specific fluorophore. At the approximate location of line 231, I would point out that F_{out} in Fig 3d not only shows the typical "defocus ring" around the point-like structures as is seen in the simulations, but also seemingly structured information in different image locations. I assume much of this signal is autofluorescence.

In the same vein, "background" in e.g. line 411 should read "STED excitation background".

Thanks for this comment.

Regarding autofluorescence, we added a few sentences to the manuscript to address the issue:

"Importantly, F_{out} also seems to contain structured features not directly related to the periphery ring of individual foci. This additional signal can therefore be associated to autofluorescence or out-of-focus foci. In either case, the fact that M-STED is able to separate this contribution from the final SPLIT image strongly supports its application to intact cells."

Line 411 was also corrected and now reads: "Finally, we exploited the subtraction of STED-induced background to perform..."

- The sentences "In order to test this idea, we simulated STED As expected, ... are indeed smeared" (Lines 291-293) are weirdly put. Observations in simulations confirm the validity of the model, and should not be discussed as if observing real data. This is what Fig 5 is for. "As expected" should be "Consequently" and "indeed" should be removed.

We thank the reviewer for this comment. The aforementioned sentences were replaced by: "In order to test this idea, we simulated STED image stacks of nuclear foci using this modified version of M-STED (Figure 4f-j). In this simulation, the final frames of the stack, that should contain better resolved foci, are smeared by the STED-induced background that counteracts the effect of the STED modulation, as seen in Figure 4f."

- The "bending towards the origin" (Line 294) is not sufficiently explained as it is not congruent with the depiction in Fig 4d.

We thank the reviewer for this comment. We added an additional explanation regarding the shape of the phase in Figure 4g:

"In the phasor plot, the presence of STED-induced background is revealed by a phasor bending towards the origin (Figure 4g). Note that the different shape of the phasor (when compared with Figure 1f) results from the fact that, in this case, the fluorescence arising from each pixel is represented in the phasor space as a combination of three different phasor positions (\mathbf{P}''_{in} , \mathbf{P}''_{out} and \mathbf{P}''_{bkgd}) as depicted in Figure 4e."

Note also that the phasor plot in the presence of STED-induced excitation (background) should be compared with the scheme reported in Figure 4e. This is because (in analogy to ref.28) we move the origin of the phasor plot to the position of the phasor of the STED-induced background (\mathbf{P}'_{bkgd}).

- The removal of STED-induced dye excitation, resulting in background emission requires the additional modulation of the excitation power. The adopter of the method should have a means to judge if this difficulty is big enough to justify the increase increase in experimental complexity. Fig 5 demonstrates a case in which this is necessary, but it would be helpful if the simulation in Fig 4 can be expanded (Suppl. Information) to include the "conventional" M-STED analysis, i.e. ignoring the presence of background. At what levels of background is a switch to the more complicated method justified?

I understand that, in the end, hopefully, producers of STED equipment will implement M-STED in its two implementations as this is only a matter of software control over the settings during an acquisition series, so that the "experimental complexity" reduces to the press of a button, but a guideline toward yes-or-no worrying about STED excitation in nevertheless warranted.

We thank the reviewer for this extremely useful comment. We performed simulations of M-STED at different level of STED-induced excitation. In the conditions of the simulations, it seems that up to a level of background of 10% the background can be ignored.

We now added Supplementary Fig.9 and citing text:

"According to simulated M-STED data, the negative effect of the STED-induced background is significant when this background level is above 10% (Supplementary Fig.9).

- The authors should consider including in the main body of the manuscript, the explanation on the unmixing of "in" and "out" information in phasor space (the SPLIT method) as is explained in the methods section (Lines 637-640). This is, the position of points along the connecting line between the "in" and "out" positions represent their mixing ratios, allowing the extraction of the fraction of "in" intensity in the pixel cloud.

We thank the reviewer for this comment. We added this explanation in Results:

“For any given pixel (x,y) , the fractions of the intensity corresponding to the center and to the periphery are obtained from the distance between the phasor measured in that pixel, $\mathbf{P}(x,y)$, and the phasor \mathbf{P}_{out} and \mathbf{P}_{in} , respectively” and also added Supplementary Figure 3 for more clarity.

- The legends for the figures sometime repeat information given in the main body of the text and can be shortened.

We thank the reviewer for this suggestion. Most legends were shortened by removing every nonessential information.

REVIEWERS' COMMENTS:

Reviewer #1 (Remarks to the Author):

The authors have well commented on all of my (and in my opinion the other referees') concerns and accordingly revised the manuscript. It reads very well. I suggest publication of this excellent piece of work as is.

Reviewer #2 (Remarks to the Author):

The authors have addressed by concerns, I have no further comments.

Reviewer #3 (Remarks to the Author):

The manuscript has benefited from the changes made. In particular, the addition of the analysis with a lower number of images, the analysis of STED background and the rearrangements in the text are helpful. My additional questions have been answered.